# Antibiotics from Insect-Associated Actinobacteria

**DOI:** 10.3390/biology11111676

**Published:** 2022-11-18

**Authors:** Anna A. Baranova, Yuliya V. Zakalyukina, Anna A. Ovcharenko, Vladimir A. Korshun, Anton P. Tyurin

**Affiliations:** 1Shemyakin-Ovchinnikov Institute of Bioorganic Chemistry, Miklukho-Maklaya 16/10, 117997 Moscow, Russia; 2Gause Institute of New Antibiotics, Bol’shaya Pirogovskaya 11, 119021 Moscow, Russia; 3Department of Soil Science, Lomonosov Moscow State University, Leninskie Gory 1-12, 119991 Moscow, Russia; 4Higher Chemical College RAS, Mendeleev University of Chemical Technology of Russia, Miusskaya sq. 9, 125047 Moscow, Russia

**Keywords:** antibiotics, chemical ecology, actinobacteria, insect microbiome, microbe–host interaction

## Abstract

**Simple Summary:**

Actinobacteria remain a key source for antibiotic discovery, and the current antimicrobial resistance crisis is becoming a driving force for actinobacteria research. Insect-associated actinomycetes are an underexplored ecological niche with prospects for the search for novel antimicrobial compounds. The described associations of leaf-cutter ants and *Pseudonocardia* bacteria or solitary wasps and *Streptomyces* bacteria were the first examples of mutually beneficial coexistence of insects with actinobacteria. On the molecular level, these systems are regulated by antibiotics. The complex relationships between insects and actinobacteria mediated by antibiotics could be important for the stability of ecosystems and agricultural production.

**Abstract:**

Actinobacteria are involved into multilateral relationships between insects, their food sources, infectious agents, etc. Antibiotics and related natural products play an essential role in such systems. The literature from the January 2016–August 2022 period devoted to insect-associated actinomycetes with antagonistic and/or enzyme-inhibiting activity was selected. Recent progress in multidisciplinary studies of insect–actinobacterial interactions mediated by antibiotics is summarized and discussed.

## 1. Introduction

Insects are the most widespread animals on our planet, playing an essential role in different ecosystems. Pollination, plant biomass destruction and the spread of tick-borne encephalitis and malaria—all these processes are impossible without insects. According to a rough evaluation by Bar-On et al. [1], about 50% of animal biomass on Earth is comprised of arthropods. Approximately one sixth of this huge number is insects (0.2 Gt of 1.2 Gt) [2].

In spite of such ubiquity and diversity, a dangerous trend in insect populations has been detected. In the 2010s, numerous studies reporting significant declines in insect populations emerged. This “decline crisis” is associated mostly with a reduction in numbers, although, in some cases, entire species have become extinct [3,4,5,6]. Globally, terrestrial insects are declining by about 11% per decade, while freshwater insects are increasing by 12% per decade, according to a meta-analysis published in *Science* [7]. However, this relatively fast change may threaten the stability of ecosystems and agricultural production.

The factors influencing insect survival should be carefully analyzed, including their interaction with microorganisms. Symbiotic bacterial species strongly associated with insects, such as *Wolbachia* sp., have a long history of study and a great potential for the regulation of insect populations [8]. Coexistence of the macro- and microorganisms is possible through close interaction and intensive signal exchange.

Chemical signaling is a very important factor for insects. Low-molecular compounds mediate insect interactions at various levels, from the individual level, where hormones control development and reproduction, to the interspecies level, where specific metabolites act as defense agents against predators [9,10,11,12]. Unsurprisingly, actinomycetes—one of the most fruitful producers of diverse secondary metabolites [13,14,15]—can interact with insects by biosynthesis of small molecules.

For example, Noodwell and co-workers [16] showed that small concentrations of volatile terpenes produced by streptomycetes attract *Drosophila melanogaster*. Fruit flies preferentially deposit their eggs on bacteria-contaminated food sources. As a result, the larvae are killed by antibiotics (cosmomycin or avermectin), which are produced by streptomycetes. This “toxic snare” could seriously affect insect populations. Nevertheless, this mechanism of ecological interaction has only recently been established. A similar attractive effect of volatile terpenes was observed for red imported fire ants [17], but, in that case, no toxic effects were detected. On the contrary, the authors suggest that the choice of *Streptomyces*- and *Nocardiopsis*-rich environments reduces the mortality of young ants from entomopathogenic fungi.

Actinobacteria remain a key source for antibiotic discovery [13], and the current antimicrobial resistance crisis is becoming a driving force for actinobacteria research. However, insect-associated actinomycetes are not just an underexplored ecological niche with prospects for the search for novel antimicrobial scaffolds [18]; they could also be important for understanding the complex relationships between different types of organisms which define the actual state and further development of ecosystems.

## 2. Methodology

In this account we make an attempt to summarize recent progress in the study of molecular ecology of insect-actinobacterial interactions mediated by antibiotics.

The search and preliminary selection of sources was performed using a Web of Science, Google Scholar, and PubMed search for keyword combinations (“insect-associated”, “actinobacteria/actinomycete”, “antibiotic/antimicrobial/inhibition”). The inclusion criteria were:

The actinobacteria (Phylum: Actinomycetota) have been isolated directly from insects (Class: Insecta) (whole animals, cuticle, gut or other internal/external organs) or from freshly sampled secreted substances. Ant nest fragments and other environmental samples were excluded;The antibiotic compounds (i.e., having some antagonistic or enzyme-inhibition activity) produced by actinomycetes have been described or, at least, antagonistic action was confirmed by bioassay;Only regular articles, reviews and book chapters were included. Patents, conference materials and preprints were excluded.

It should be noted that the insect-associated microbiome as a source of novel natural products was a subject of several other reviews. The discussion of insect–actinobacterial symbiotic relationships and their relevance for drug discovery started from comments by M. Kaltenpoth [19] and H. B. Bode [20]. During the next decade (2010s), this topic was highlighted and/or reviewed in several works [3,21,22,23,24,25,26,27].

The latest systematic review on natural products from insect-associated microorganisms was published by C. Beemelmanns et al. [28] in 2016. Despite it not being focused on a specific genus and type of compounds, we could recognize it as a starting point for our literature selection: the current review is covering January 2016–August 2022. From the latest years we should highlight some works on specific types of insects or ecological niches: fungus-growing ants [29], African edible insects [30], neotropical insects [31] and fungus-farming termites [32]. A critical review on microbial symbionts of insects as a source of new antimicrobials from our Belgian colleagues [18] is the most relevant and closest to our work, but it does not have a systematic character and is not focused on actinomycetes as a major source of antimicrobials.

Using the above-mentioned criteria and methodology, we identified 71 research papers from 2016 [26,33,34,35,36,37,38,39,40,41,42,43,44,45,46,47,48,49,50,51,52,53,54,55,56,57,58,59,60,61,62,63,64,65,66,67,68,69,70,71,72,73,74,75,76,77,78,79,80,81,82,83,84,85,86,87,88,89,90,91,92,93,94,95,96,97,98,99,100,101,102].

A total of 14 other works were excluded due to the source of actinobacteria isolation (insect-associated environments such as dung beetle’s brood ball, wasp nest, termite fungus comb, bee pollen, etc.—see Appendix A). The most important information from the selected materials is summarized in Appendix A. Here, we present our comments and conclusions based on analysis of the data.

## 3. General Remarks: Data Pre-Processing

All the presented studies have the same experimental design, which can be illustrated by Figure 1:

First of all, in some works (11 out of 71), this workflow is not complete: no compounds were isolated and characterized, only antagonistic activity of actinomycete strains was detected. In others, we could find repeated data in strain descriptions (marked as “previously described”, 10 out of 60), which means that new compounds were isolated from the same strains. This could be interpreted as evidence for the high biosynthetic potential of actinobacteria. Other missing values (marked as “no data”) are caused by an absence of experimental details in the original works.

## 4. Data Analysis: Main Trends

The geographical distribution of sample collection locations is depicted on Figure 1. We could see a clustering of sampling locations—samples were often collected in compact regions: Central America, South Brazil, South Africa, East China, North-East China and South Korea. On the contrary, it is easy to see that most of the world area is not involved in insect–actinobacteria symbiosis studies.

The collected insects could be classified into major taxonomic categories (order, family) as shown in Figure 2. The most studied insect phylum, by a wide margin, was ants (Hymenoptera: Formicidae). We also have to note termites (Blattodea: Termitidae and Rhinotermitidae), bees (Hymenoptera: Apidae), Silphidae beetles (Coleoptera: Silphidae) and grasshoppers (Orthoptera: Acrididae). Unfortunately, in three cases [36,61,96], the taxonomic status of the insects was not determined.

Only the classical isolation approach was used in the selected papers. The most used isolation media were chitin agar (C/N-source—chitin) and Gause agar №1 (C-source—soluble starch, N-source—ammonium salts) (see Appendix A). These are well-known selective media for actinobacteria isolation.

*Streptomyces* was the most isolated genus (see Figure 3); it is typical for all culture-dependent studies of actinobacteria. From non-streptomycete actinobacterial genera, the *Pseudonocardia*—a well-studied symbiont of Attini ants—was the most mentioned. New species (8, from which 2 are synonymic) were described: 6 *Streptomyces*, 1 *Amycolatopsis* and 1 from genus *Actinomadura*.

Here, we grouped the isolated compounds by their biosynthetic origin: peptides, polyketides, alkaloids and other. Class and structure assignment is summarized in Appendix A. For all schemes (here and in Appendix A), new elucidated structures are colored black, while known metabolites are colored gray. The main statistics on the isolated metabolites are presented in Figure 4.

Polyketides are the leading class among described antibiotics (about half of the total). However, more importantly, structural novelty is quite high for all main classes of compounds.

Despite a significant number of publications and detected metabolites, the ecology of the isolated compounds is discussed in just 15 out of 71 papers. In all cases, it is protective symbiosis, but these symbiotic relationships could be divided into two main types: protection of food sources (for fungus-growing insects) and direct protection against entomopatogenic fungi and bacteria (Figure 5).

Food source protection is described extensively for fungus-growing termites and ants (Attini tribe). The fungal cultivar is an important food source for these social insects with specialized diets, and the presence of pathogenic fungal species (*Escovopsis* sp. [103] for ants and *Pseudoxylaria* sp. for termites [32]) is a critical factor for colony survival. Compounds selectively inhibiting fungal pathogens and other non-specialized pathogens while being neutral to fungal cultivar are widely represented in the biosynthetic arsenal of actinobacteria.

Examples of antibiotics used by insects for food source protection are summarized on Figure 2. The majority of the structures are associated with leaf-cutter ants (and their actinobacterial symbionts). We should highlight some newly described antibiotics: an unusual non-polyene macrolide [104] cyphomycin (**77**) [44,69] and peptide antibiotics attinimicin (**23**) [38] and dentigerumycin F (**8**) [48].

Another fruitful source for isolation of the above type of antibiotics is termites. Polyketides macrotermicin A (**40**) and C (**47**) from termite-associated *Amycolatopsis* sp. M39 [90] active against parasitic *Pseudoxylaria* sp. Polyenes (**74**) and (**75**) were isolated from termite-associated *Streptomyces* sp. HF10 [42]. Both compounds have higher activity against pathogenic fungi (*Xylaria* sp. and *Metarhizium anisopliae*) than cultivar *Termitomyces*.

Only one example of food source protection is described for a different type of insects: bacteria *Streptomyces griseus* XylebKG-1 isolated from bark beetle *Xyleborus saxesenii* produce cycloheximide (**81**). This well-known antifungal inhibits the growth of parasitic fungus *Nectria* sp., but not of mutualistic *Raffaelea sulphurea*.

However, the list of such substances could not be complete without well-known inorganic compounds: sulphur and ammonia. Elemental sulfur (S_8_) produced by *Streptomyces chartreusis* strain ICBG323 [64] had antifungal activity against *Escovopsis* sp. *Streptomyces* sp. Av25_4, and other isolated actinobacterial species overproduce ammonia (up to 8 mM), which completely inhibits the growth of *Escovopsis weberi* due to strong basic pH [39]. These findings clearly indicate that not just complex antibiotics, but also simple inorganics excreted by symbiotic bacteria, are important for insect ecology.

Antibiotics utilized by insects as self-protection agents are shown in Figure 3.

Well-known cytotoxic dipeptide antimycin A (**19**) was isolated from ant-associated *Streptomyces albidoflavus* A10. At low concentrations, it inhibits the growth of entomopathogenic fungi (*Conidiobolus coronatus*, *Beauveria bassiana* and others) [51]. Ionophore antibiotics (**84–88**) isolated from wasp-associated *Streptomyces* sp. M54 were effective against *Hirsutella citriformis*. This fungus is a natural enemy of the host—the wasp *Polybia plebeja* [45]. Peptides meliponamycins (**15**, **16**) [56], glycosylated antibiotics lobophorins (**152**–**153**) and polycyclic antibiotics (**164**–**165**) [65] were isolated from bee-associated actinobacteria. All these compounds exhibited high activity against a bee-specialized pathogen—*Paenibacillus larvae*, the causative agent of American Foulbrood.

Thiopeptide antibiotic GE37468 (**17**) (Figure 4) from *Pseudonocardia* sp., associated with ant *Trachymyrmex septentrionalis*, was described as a selective inhibitor of other ant-associated *Pseudonocardia* strains [54]. Its niche-defense role is an interesting phenomenon which opens up a new dimension in insect–actinobacteria symbiosis research.

## 5. Conclusions and Outlook

In the current review, we identified the main trends and problems of relevance to insect–actinomycete interaction research. Actinobacterial species are likely involved into multilateral relationships between insects, their food sources, infectious agents, predators, etc. Antibiotics and related natural products play an essential role in such systems.

Our brief analysis clearly indicates that we are only at the very beginning of the path leading to an understanding of the actual diversity of the ecological relationships between actinomycetes and insects. To fill the gaps in knowledge, we need more research data. First, efforts should be made to expand the range of studied insect hosts, with particular attention paid to species whose life cycle is associated with soil, plants, decaying organic debris and other substrates rich in microorganisms. It is noteworthy that various stages of the insect life cycle should be considered as distinct objects of study, since the ecology of larvae and imagines in many species is fundamentally different (lifestyle, habitat, diet etc.).

Traditional culture-dependent methods continue to be the main way of mining for active strains. The efficiency of actinobacteria isolation could be improved using innovative isolation techniques such ichip [105]. At the same time, the study of the metagenome of microbial communities associated with insects can provide a significant amount of valuable information. However, active antibiotic producers are often minor constituents of communities and their role may be underestimated.

Understanding insect–actinobacteria systems at the molecular level requires complex, multidisciplinary approaches. Proper chemical characterization of active metabolites and appropriate bioactivity assays should become the base of ecological research in this area. Insect ecology remains unclear without works at the intersection of microbiology, chemistry and animal studies.

We have found that the potential of this area is much greater than just an exotic source of compounds. The creation of new methods and agents for the prevention of insect-borne infectious diseases, pest control and other related areas may be stimulated by the study of insect-associated actinomycetes and their secondary metabolites.

## Data Availability

All datasets generated or analyzed during the current study are openly available in the Appendix A.

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
