# Peer review of "Antibiotics from Insect-Associated Actinobacteria"

_biology, 2022, doi:10.3390/biology11111676_

Round 1
Reviewer 1 Report
The manuscript Baranova et al. is a revision of the state of the art in the field of actinobacteria associated with insects and the antibiotics produced by these bacteria.
The review is sound, well structured and well written. It compiles the papers from 2016 to date and makes a detailed analysis of the information contained in those articles. Therefore, it is a good support for future investigations in the same thematic area.
The methodology is explained clearly and the figures and tables are appropiate.
I consider the paper suitable to be published just with two minor revisions, just as follows:
-In the molecular schemes legends included the information of what grey and black means (known metabolites and new elucidates ones respectively.
-Revised the italics in the microorganism and insect names in the references. Some of the are missing. Besides, in the reference 18 the year must be in bold.
Author Response
Response to Reviewer 1 Comments
Point 1: In the molecular schemes legends included the information of what grey and black means (known metabolites and new elucidates ones respectively.
Authors’ response: Thanks for your careful reading. The corrections were included into the revised version. All corrections are highlighted yellow.
Point 2: Revised the italics in the microorganism and insect names in the references. Some of the are missing. Besides, in the reference 18 the year must be in bold.
Authors’ response: The references section was checked both in manuscript and supplementary materials. All names, years and journal abbreviations were corrected.

Reviewer 2 Report
Reviewers Report on: biology-2024921
Title: Antibiotics from insect-associated actinobacteria
The main findings of the study.
This paper reviews recent (from 2016) research papers on antibiotic-producing actinobacteria from a specific source - insects.
General remarks:
The review presents an interesting analysis of recent literature on the topic. The authors made a comprehensive analysis but managed to maintain a very readable article on this exciting and underexplored field. It is informative but not overloaded with information. It is not necessary to further emphasize the need to discover new antibiotics. In addition, understanding the role of antibiotics in interspecies interactions is very important, not only from the aspect of ecology, specifically insects but as a model that, if understood, can be manipulated for the good of insects and people.
I have no substantive objections. However, I feel that the authors didn't emphasize enough the knowledge gaps and how they can be filed. Especially in the Conclusion section, recommendations for improving the isolation process of actinobacteria from insects to increase the diversity of isolates and, consequently, increase the chance of discovering new antibiotics, is missing. Similarly, to give some ideas on what to do to improve the study of the role of antibiotics in interaction, etc. For every aspect of studying insect-associated antibiotic-producing actinobacteria that you covered, you should give some thoughts on improving research and filling the gaps.
Minor points:
The paper could benefit from light English editing.
Paragraph (L51-56) calls for literature citing.
Fig 3A is not informative; you can omit it. In Fig 3B, match the color of the pie sections and the squares where you put the newly described species.
Information in two paragraphs (L180-188) seems like outside addition that doesn't really fit. It needs to be better incorporated, and allow yourself to elaborate more on this.
In Schemes 2, 3, and 4, please, emphasize in the image caption which compounds are given in gray and which in black.
Author Response
Response to Reviewer 2 Comments
Point 1: I have no substantive objections. However, I feel that the authors didn't emphasize enough the knowledge gaps and how they can be filed. Especially in the Conclusion section, recommendations for improving the isolation process of actinobacteria from insects to increase the diversity of isolates and, consequently, increase the chance of discovering new antibiotics, is missing. Similarly, to give some ideas on what to do to improve the study of the role of antibiotics in interaction, etc. For every aspect of studying insect-associated antibiotic-producing actinobacteria that you covered, you should give some thoughts on improving research and filling the gaps.
Authors’ response: The “Conclusions” section was renamed as “Conclusions and outlook” and intensively rewritten according the Reviewer’s recommendations.
Point 2: The paper could benefit from light English editing.
Authors’ response: The corrections were included into the revised version. All corrections are highlighted yellow.
Point 3: Paragraph (L51-56) calls for literature citing.
Authors’ response: References 9-15 were added into revised version of the manuscript.
Point 4: Fig 3A is not informative; you can omit it. In Fig 3B, match the color of the pie sections and the squares where you put the newly described species.
Authors’ response: Figure 3A was transferred into supplementary materials (now Figure S1). Figure 3B was improved (now Figure 3).
Point 5: Information in two paragraphs (L180-188) seems like outside addition that doesn't really fit. It needs to be better incorporated, and allow yourself to elaborate more on this.
Authors’ response: The mentioned paragraphs were rewritten (now lines 180-190).
Point 6: In Schemes 2, 3, and 4, please, emphasize in the image caption which compounds are given in gray and which in black.
Authors’ response: Thanks for your careful reading. The captions were improved.
